# Latent class analysis of barriers to HIV testing services and associations with sexual behaviour and HIV status among adolescents and young adults in Nigeria

Okikiolu Badejo[1,2,3]*, Edwin Wouters[2], Sara Van Belle[1], Anne Buve[1], Tom Smekens[1], Plang Jwanle[3], Marie Laga[1], Christiana Nöstlinger[1]

1 Department of Public Health, Institute of Tropical Medicine, Antwerp, Belgium, 2 Department of Sociology, University of Antwerp, Antwerp, Belgium, 3 APIN Public Health Institute, Abuja, Nigeria

* Okikolubadejo@gmail.com

**Data Availability Statement:** All data files are available from the website of the Nigeria AIDS Indicator and Impact Survey (NAIIS) obtainable via

## Abstract

### Introduction

Adolescents and young adults (AYA) face multiple barriers to accessing healthcare services, which can interact, creating complex needs that often impact health behaviours, leading to increased vulnerability to HIV. We aimed to identify distinct AYA subgroups based on patterns of barriers to HIV testing services and assess the association between these barrier patterns and sexual behaviour, socio-demographics, and HIV status.

### Methods

Data were from Nigeria's AIDS Indicator and Impact Survey (NAIIS, 2018) and included 18,612 sexually active AYA aged 15–24 years who had never been tested for HIV and reported barriers to accessing HIV testing services. A Latent class analysis (LCA) model was built from 12 self-reported barrier types to identify distinct subgroups of AYA based on barrier patterns. Latent class regressions (LCR) were conducted to compare the socio-demographics, sexual behaviour, and HIV status across identified AYA subgroups. Sex behaviour characteristics include intergenerational sex, transactional sex, multiple sex partners, condom use, and knowledge of partner's HIV status.

### Results

Our LCA model identified four distinct AYA subgroups termed 'low-risk perception' (n = 7,361; 39.5%), 'consent and proximity' (n = 5,163; 27.74%), 'testing site' (n = 4,996; 26.84%), and 'cost and logistics' (n = 1,092; 5.87%). Compared to adolescents and young adults (AYA) in the low-risk perception class, those in the consent and proximity class were more likely to report engaging in intergenerational sex (aOR 1.17, 95% CI 1.02–1.35), transactional sex (aOR 1.50, 95% CI 1.23–1.84), and have multiple sex partners (aOR 1.75, 95% CI 1.39–2.20), while being less likely to report condom use (aOR 0.79, 95% CI 0.63–0.99). AYA in the testing site class were more likely to report intergenerational sex (aOR 1.21, 95%

the Federal Ministry of Health. Contact information for data access areas follows: 1. nada@naiis.ng 2. osajayabs@gmail.com The two email addresses are affiliated with both the Federal Ministry of Health and the Nigeria AIDS Indicator and Impact Survey (NAIIS). This is also to confirm that neither of the above email addresses belongs to an author affiliated with this manuscript.

**Funding:** OB was supported with research funding from the Belgium Directorate-General for Development Cooperation (DGD) awarded through the Institute of Tropical Medicine Antwerp, Belgium. SVB received funding from the Flanders Research Foundation (FWO) grant number 1221821N. The funders had no role in study design, data collection and analysis, decision to publish, or preparation of the manuscript.

**Competing interests:** No competing interests declared

CI 1.04–1.39) and transactional sex (aOR 1.53, 95% CI 1.26–1.85). AYA in the cost and logistics class were more likely to engage in transactional sex (aOR 2.12, 95% CI 1.58–2.84) and less likely to report condom use (aOR 0.58, 95% CI 0.34–0.98). There was no significant relationship between barrier subgroup membership and HIV status. However, being female, aged 15–24 years, married or cohabiting, residing in the Southsouth zone, and of Christian religion increased the likelihood of being HIV infected.

## Conclusions

Patterns of barriers to HIV testing are linked with differences in sexual behaviour and socio-demographic profiles among AYA, with the latter driving differences in HIV status. Findings can improve combination healthcare packages aimed at simultaneously addressing multiple barriers and determinants of vulnerability to HIV among AYA.

## Introduction

The global goal to end the AIDS epidemic by 2030 requires expanded HIV prevention and treatment interventions, making them accessible to all in need [1, 2]. However, adolescents and young adults (AYA), particularly in low and middle-income countries, face challenges in progressing toward this goal [3]. These challenges have led to significant disparities in achieving the global 95-95-95 targets for HIV elimination [1, 4]. The 95-95-95 targets are a set of global goals for HIV prevention and treatment that were set by the Joint United Nations Programme on HIV/AIDS (UNAIDS) in 2020. The goals are aimed for 95% of all people living with HIV to know their HIV status, 95% of all people with diagnosed HIV infection to receive sustained antiretroviral therapy, and 95% of all people receiving antiretroviral therapy to have viral suppression by 2025 [1]. As of 2020, treatment coverage among young adults living with HIV aged 15–24 was estimated at 55%, significantly lower than the 75% coverage among those over 25 years [1]. In 2019, the number of new HIV infections among adolescents and young adults aged 15 to 24 decreased to an estimated 460,000 new infections, representing a 46% decline since the year 2000. However, this is still eight times higher than the global target of fewer than 50,000 new infections by 2025 [5, 6]. Remarkably, over 80% of these new infections occurred in sub-Saharan Africa [5]. Nigeria, one of the countries with the highest burdens of adolescents living with HIV (ALHIV) in sub-Saharan Africa, continues to experience increases in mortality among both younger and older adolescents in contrast to other countries in the region [7, 8]. In addition, only 31% of young people aged 15–24 in Nigeria are aware of their HIV status, significantly below the national average of 46.9% [9].

The slower progress in AYA HIV-related outcomes is linked to health services' inability to address young peoples' multiple and complex needs [10, 11]. These needs encompass a range of services and are compounded by multiple barriers to access [10, 11]. These barriers include financial, social (such as stigma), and informational obstacles [12, 13]. In Nigeria, there is a noticeable lack of services related to mental health, reproductive health, clinic transitional care, and psychosocial support for young people living with HIV which might explain the poor health-seeking behaviour and heightened vulnerability to HIV observed in this age group [14–20]. Despite the need for comprehensive and integrated approaches to address these multifaceted needs, healthcare services often fall short due to fragmented and siloed care provision which presents multiple barriers to access [12]. Recognising this challenge, there has been a

growing trend toward bundled healthcare interventions [21, 22] that combine various services and support to comprehensively and simultaneously address young people's multiple barriers and needs. One such combination of health service interventions in Nigeria is the "Minimum package for Youth-friendly services" targeting young people [23]. While bundled healthcare interventions are considered effective when they simultaneously address social, behavioural and structural barriers, their outcomes in practice have been mixed [19–26]. To enhance their effectiveness, a more targeted and holistic approach that aligns with specific patterns of barrier combinations faced by young people with diverse challenges is needed [19–26].

To address this need, this study utilises Latent Class Analysis (LCA) to model barrier patterns related to HIV testing services among AYA in Nigeria. LCA is a statistical modelling technique that identifies distinct subgroups (latent classes) within a population based on shared patterns of specific characteristics [26]. It classifies individuals into unobserved subgroups, estimating these based on multivariate clustering of observed variables to account for population heterogeneity [27, 28]. The number of subgroups in LCA is not determined a priori; instead, it is selected based on a combination of different model-fit criteria. LCA offers several advantages over similar cluster analysis techniques, such as k-means. It relies more on formal criteria to determine the final model and is flexible in accommodating different variable scales [26]. Furthermore, LCA addresses some methodological challenges encountered in traditional subgroup analysis, such as high type 1 error, low statistical power, issues related to collinearity, and difficulties analysing and modelling complex and multidimensional interactions between observed variables [26, 29, 30]. LCA can be extended by incorporating external covariates and outcomes to enhance understanding of the relationships between latent classes and external variables [31]. This is achieved through Latent class regression (LCR), where external covariates or outcomes are regressed on latent classes or subgroups [32]. LCA has been previously employed in studies focused on sexual health and HIV in young people, as well as barriers to health services [19, 33–35]. However, to the best of our knowledge, the application of LCA and LCR to investigate association between barriers to HIV testing services, sexual behaviour and HIV status in young people has not been previously explored. This study builds upon previous studies in Nigeria investigating the relationship between the utilisation of HIV testing and prevention services, sexual behaviour and vulnerability to HIV in AYA in Nigeria [36–38].

Our study aims to: identify latent classes of AYA based on shared patterns of barriers to HIV testing services using LCA, identify and compare the sexual and sociodemographic characteristics of AYA across different latent classes using LCR, and explore the association between latent class membership of AYA and observed HIV status using LCR, adjusting for sexual and sociodemographic characteristics.

## Method

### Context

Nigeria is administratively divided into six geopolitical zones (Northwest, Northeast, North-central, Southwest, Southeast, and Southsouth) comprising 36 states and one Federal capital (Fig 1). Despite economic growth in recent decades, the country continues to face high poverty rates [39]. Regional disparities are significant, with poverty levels ranging from 30% in the South to over 60% in the North [39]. Certain regions in the North have poverty rates exceeding 80% as of 2020 [39]. These disparities extend to health and education indices.

Women in the North have lower educational attainment, with average years of formal education reaching less than half of that in the South [39]. Similarly, households in the North are less asset-wealthy and have experienced increasing poverty compared to households in the

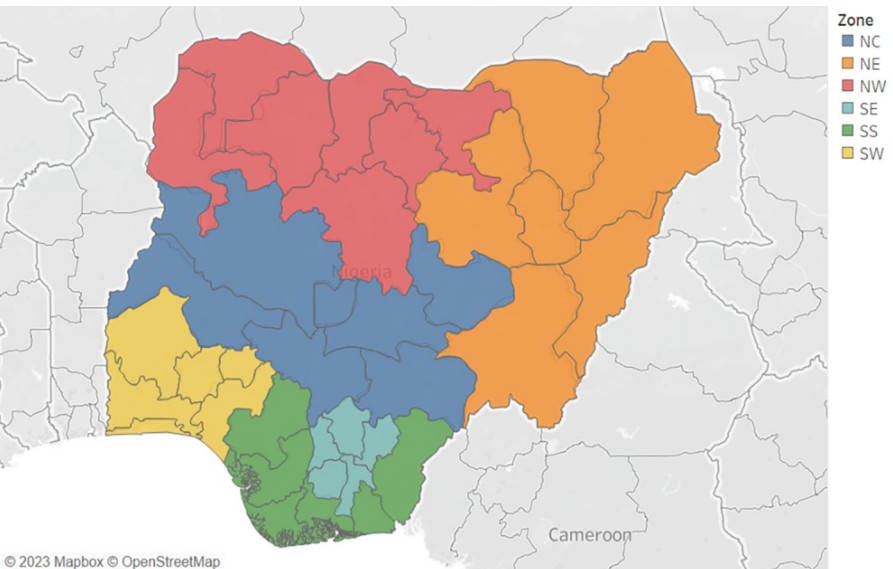

**Fig 1. Map of Nigeria by zones.** Map data available from ©OpenStreetMap under the Open Database License.

South [39]. Maternal healthcare utilisation has been consistently lowest in the Northwest and Northeast regions and, together with the North Central, have the highest rates of child and under-five mortality [40, 41].

## Data

This study used data from Nigeria's National AIDS Indicator and Impact Survey (NAIIS). NAIIS was a nationally representative, cross-sectional, two-stage, population-based survey of households. NAIIS used a two-stage cluster sampling technique, selecting enumeration areas (EAs) followed by households. The sampling frame consisted of 662,855 EAs, 28,900,478 households and 140,431,798 persons based on the 2006 Census, with an average number of households and persons per EA of 44 and 212, respectively. The eligible survey population included adults aged 18–64 years, emancipated minors aged 15–17 years, children and adolescents aged 10–14 years, and children aged <10 years.

Participants' recruitment, data and blood sample collection occurred from July 2018 to December 2018, focusing on HIV and related health indicators, including hepatitis B virus (HBV) infection, hepatitis C virus (HCV) infection, HBV/HIV co-infection and HCV/HIV co-infection [9]. For adults aged 15–64, the interview response rate was 91.6% for women and 88.2% for men; the blood draw response rate was 92.9% for women and 93.6% for men. For adolescents aged 10–14, the interview response rate was 86.8% for women and 86.2% for men, and the blood draw response rate was 91.2% for women and 92.3% for men. NAIIS is the first survey in Nigeria to estimate national HIV incidence and viral load suppression (VLS).

Three types of questionnaires were used: household questionnaire, adolescent questionnaire for individuals aged 10–14 years, and adult questionnaire for women and men aged 15 years or older. The adolescent and adult questionnaires collected information from eligible adolescents aged 10–14 years and adults aged 15 years and older on basic demographic characteristics, marriage, sexual activity, HIV and STI knowledge, attitudes and behaviours, and previous HIV testing. In addition to the interview, blood was drawn from consenting participants for HIV antibody testing. Final HIV status was determined using rapid HIV testing and

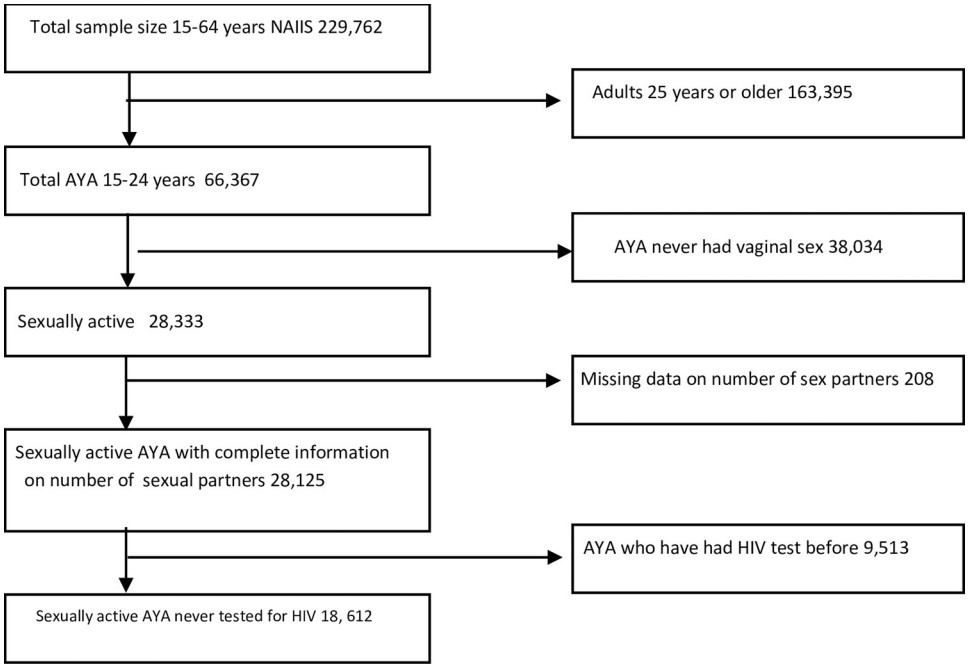

**Fig 2. Study flowchart showing inclusion and exclusion criteria.**

Geenius™ HIV 1/2 confirmatory testing on all reactive rapid test results. The testing procedures and national testing algorithm used have been described elsewhere [9] and summarised in S1 File. Personal identifiers were excluded from the data set before analyses were performed. Details of the survey methods and questionnaire are available on the study website: https://nadanaiis.nascp.gov.ng/home.

## Study participants

Our inclusion criteria, as illustrated in Fig 2, include adolescents (15–19 years) and young adults (20–24 years); being sexually active (that is, have had vaginal sex before the survey); never having been tested for HIV before the survey. We excluded participants who had missing information on the number of sexual partners.

## Measures

Study measures were based on the adult questionnaire. The questionnaire has ten modules focused on respondent consent and background and sociodemographic characteristics, marriage, reproduction, children, male circumcision, sexual activity, HIV testing, HIV care and treatment, tuberculosis and other health issues, and gender norms. Notably, NAIIS 2018 used only vaginal sex to determine being sexually active. We also retrieved the results of blood testing for HIV biomarkers. Details of the study measures are provided in Table 1.

## Statistical analysis

For our first objective, we used LCA to identify latent classes or subgroups of AYA based on shared combination patterns of the 12 manifest barrier variables listed in Table 1, supported by the literature as relevant to HIV testing uptake in Nigeria and other settings [18, 36]. The LCA consisted of the following steps:

**Table 1.  Study measures.**

| Variable | Measure description | Recode |
|---|---|---|
| **Socio-demographics** | | |
| Age | Continuous; based on the question "How old were you on your last birthday?" | Binary; based on a recode of age into two categories: 15-19/20-24 years |
| Sex | Binary; based on the questionnaire item "Item check: Is respondent male or female?" | Not recoded |
| Religion | Nominal: based on the question "What is your religion?" Seven response options: "Islam", "Christianity", "Traditional", "No religion", "Other", "Don't know", "Refused" | Recoded into three categories: "Islam", "Christianity", and "Others." |
| Marital status | Nominal; based on two questions: "Have you ever been married or lived together with a [man/woman] as if married?" and "What is your marital status now?". Seven response options: "Married", "Living together","Widowed", "Divorced", "Separated", "Don't know", "Refused" | Recoded into four response categories: "Never married", "Married or living together", 'Divorced or separated', "Widowed." |
| Education level | Ordinal: based on two questions: "Are you currently enrolled in school?" and"What is the highest level of school you have attended? "No education", "Primary", "Secondary", "Tertiary", "Other" | Not recoded |
| Wealth quintile | Ordinal with five categories: "Lowest", "Second", "Middle", "Fourth", "Highest" | Not recoded |
| Type of place of residence | Binary, with two categories: "Urban" and "Rural." | Not recoded |
| Geopolitical zone | Nominal with six categories: "North-East", "North-West", "North-Central", "South-East", "South-West", "South-South" | Not recoded |
| **Sexual behaviour** | | |
| Intergenerational sex | Categorical. Computed based on respondent age and answer to "How old is [INITIALS]? Please give your best guess" regarding the last three sex partners in the last 12 months before the survey. | Three categories<br>"No"–Reported sexual partner/s is/are less than ten years older or younger than the respondent.<br>"Yes, with minor"–At least one of the respondent's reported sexual partner/s is ten or more years younger or older than the respondent, AND one of the respondent or partner is less than 18 years.<br>"Yes, with non-minor"—At least one of the respondents' reported sexual partner/s is ten or more years younger or older than the respondent, AND none of the respondents or partner is less than 18 years. |
| Condom use at last sex with non-marital, non-cohabiting partners in the past 12 months | Ordinal response: The last time you had sex with [INITIALS], was a condom used? And What is your relationship with [INITIALS]? | Three categories<br>'No sex with non-marital partner in past 12 months', 'Sex, condom used', 'Sex, condom not used' |
| Number of sexual partners in the past 12 months | Continuous based on the answer to the question: People often have sex with different people over their lifetime. In total, with how many different people have you had sex in the last 12 months? | Recoded into three categories: 0, 1, 2 or more |
| Partner HIV status known | Multinominal with seven response options to the question, What is the HIV status of (INITIALS)?<br>Options are:<br>"I think (INITIALS) is positive."<br>"(INITIALS) told me he/she is positive",<br>(INITIALS) is positive, tested together,<br>I think (INITIALS) is negative,<br>(INITIALS) told me he/she is negative,<br>(INITIALS) is negative, tested together,<br>Don't know the status,<br>Refused. | Recoded into two categories, "Yes" consisting of the following responses: (INITIALS) is positive, tested together, (INITIALS) is negative, tested together,<br>"No" consisting of the following responses ("Don't know status", "I think (INITIALS) is positive", "(INITIALS) told me he/she is positive", I think (INITIALS) is negative, (INITIALS) told me he/she is negative)<br>Refused option treated as missing data |

(*Continued*)

**Table 1.** (Continued)

| Variable | Measure description | Recode |
|---|---|---|
| Transactional sex | Nominal based on the answer to the question "Did you enter into a sexual relationship with [INITIALS] because [INITIALS] provided you with or you expected that [INITIALS] would provide you gifts, help you to pay for things, or help you in other ways?<br>Four Response categories: "Yes", "No", "Don't know", "Refused" | Recoded into two categories:<br>Yes, consisting of the following responses: "Yes."<br>No, consisting of the following responses: "No."<br>"Don't know" and "refused" options are treated as missing data. |
| **Barriers to HIV testing** | | |
| Multi-response item based on response to the question"Why have you never been tested for HIV? | **Response items:**<br>• Don't know where to test (Y/N)<br>• Test costs too much (Y/N)<br>• Transport costs too much (Y/N)<br>• Too far away (Y/N)<br>• Afraid others will know about the results (Y/N)<br>• Don't need test/low risk (Y/N)<br>• Did not receive permission from spouse/family (Y/N)<br>• Afraid spouse/partner/family will know results (Y/N)<br>• Don't want to know I have HIV (Y/N)<br>• Cannot get treatment for HIV (Y/N)<br>• Test Kits not available (Y/N)<br>• Religious reasons (Y/N)<br>• Don't know (Y/N)<br>• Other<br>• Refused | Responses to 'Other' were provided as free-text. NAIIS survey team analysed the free-text responses and assigned each back into existing categories.<br>Refused was treated as missing data |
| **Final HIV status** | | |
| | For participants aged 18 months, the algorithm for classification of final HIV status is determined using blood rapid HIV testing and Geenius™ HIV ½ confirmatory testing on all reactive tests. In addition, Western Blot, TNA PCR and VL RNA PCR were done on discrepant results. For participants under 18 months, the algorithm for classification of final HIV status included results from rapid HIV testing and HIV TNA PCR. | Coded into "HIV negative" and"HIV positive." |

The first step included an iterative process of building models by gradually increasing the number of classes and employing various fit indicators to assess model fit. This method ensured the identification and selection of the model with the optimal number of classes. We used the following model fit indices to select the optimal model: Akaike Information Criteria (AIC), Bayesian Information Criteria (BIC), Entropy values and the Lo, Mendell and Rubin Likelihood Ratio Test (LMR). Smaller AIC and BIC values are preferable, while Entropy values should be close to 1 [26]. The LMR test indicates whether a model fits better than the model with one fewer class (the complex survey design did not permit using the Bootstrap Likelihood Ratio Test (BLRT)) [26, 30].

After selecting the model with the optimal number of classes, the next step involved assigning each AYA to a specific class. To achieve this, we employed the maximum probability rule, often referred to as the 'most likely class' approach. In this approach, individuals are allocated to the class with their highest estimated probability [42]. Although an alternative method could involve assigning individuals to multiple classes based on their probabilities for each class, we opted for the most-likely class approach. This decision allowed us to examine variations in the frequencies of sociodemographic and sexual behavior characteristics among classes. This practice is considered acceptable, especially when entropy exceeds 0.80 [27].

Following class assignment, the subsequent step involved scrutinizing the conditional response probabilities for each barrier type within the identified classes. We applied labels to each class based on the HIV testing barrier(s) exhibiting higher conditional response probabilities. To address missing data, we employed the Mplus procedure of Bolck–Croon–Hagenaars

(BCH) with multiple imputations on 20 data sets, accommodating the complexities of the survey design [28].

For the second objective, we used *LCR with covariates* [28, 43] to compare the sexual and sociodemographic characteristics of AYA across the different assigned latent classes or subgroups. The integrated 3-step approach in Mplus (R3STEP functionality) allowed us to compare these characteristics between classes while accounting for inherent potential classification errors in the most likely class approach [28, 43] as follows: (a) each AYA was first assigned using the most likely class approach, but their distributed probabilities across classes were computed and saved (b) We calculated the measurement error for each observation based on the probability used for its most likely class assignment and the probabilities distributed across different classes (c) In the regression model, we represent the latent class variable with the most likely class assignment and include pre-specified measurement errors obtained in step (b) [43]. We used the TYPE = COMPLEX MIXTURE feature to accommodate the complex survey nature of the dataset [28, 43]. To assess potential multicollinearity among the sexual and sociodemographic factors, we examined correlation matrices and calculated variance inflation factors (VIFs).

For our third objective, we used *LCR with distal outcomes* [28, 43] to explore the relationship between latent class membership of AYA and HIV positivity rates. We utilised Mplus' automatic BCH regression procedure, which accounted for potential classification errors and accommodated the complex survey nature of the dataset [28, 44]. We then adjusted for sociodemographic and sexual characteristics in the model. We present crude and adjusted odds ratios and 95% confidence intervals (95% CI). Statistical significance was determined by the 95% CI of crude and adjusted odds ratios not overlapping 1.00 and, when indicated, p value less than 0.05.

To examine the consistency of our results, we conducted a sensitivity analysis using only complete cases from the non-imputed dataset. This analysis mirrored the key steps employed for the first and third objectives, specifically, using LCA to uncover latent classes or subgroups of AYA based on their shared patterns across the 12 barrier variables detailed in Table 1; and using LCR with distal outcomes to examine the relationship between latent class membership and HIV positivity rates, adjusting for sexual and sociodemographic factors.

We used STATA version 17.0 [45] to prepare the data, including cleaning and recoding variables and checking for missing data. The cleaned dataset was exported into Mplus version 8.8 for analysis [44].

## Ethics approval and consent to participate

NAIIS 2018 had approval from the Nigeria National Health Research Ethics Committee and the IRBs of the US Center for Disease Control (CDC) University of Maryland Baltimore. All participants, per United States regulations, provided written and verbal informed consent or assent. For minors aged 10–17, written and verbal consent were obtained from parents or guardians for interviews and blood draws. Subsequently, assent was obtained from the minors.

Permission to access and use NAIIS 2018 data for this study was granted by a review committee established by the Nigeria Federal Ministry of Health. The broader study, which this study is a part of, also received ethical approvals from the Institute of Tropical Medicine, Belgium, and the APIN Public Health Initiatives, Nigeria. The approvals included waivers for secondary data analysis.

NAIIS2018 study data has been carefully anonymised to protect privacy, and we adhered to the terms and conditions outlined in the provided confidentiality agreement. Further information on the terms and conditions for NAIIS 2018 data access can be found at https://nadanaiis. nascp.gov.ng/home.

**Table 2. Model fit statistics.**

| Class | AIC | BIC | BIC (sample size adjusted) | Entropy | LMR p-value |
|---|---|---|---|---|---|
| 1 | 102031.360 | 102125.339 | 102087.204 | | |
| 2 | 95461.811 | 95657.600 | 95578.152 | 0.859 | <0.001 |
| 3 | 92908.502 | 93206.101 | 93085.340 | 0.892 | 0.0112 |
| 4 | **91718.214** | **92117.624** | **91955.549** | **0.873** | **<0.001** |
| 5 | 91327.984 | 91829.204 | 91625.816 | 0.851 | 0.1969 |
| 6 | 91035.288 | 91638.318 | 91393.616 | 0.844 | 0.1363 |

Figures in bold show the fit of the four class model selected. AIC Akaike Information Criterion, BIC Bayesian Information Criterion, LMR Lo- Mendell-Rubin likelihood ration test

## Result

We analysed 18,612 AYA who met the study inclusion criteria (Fig 2). Distribution of AYA by sociodemographic characteristics, sexual activity, barriers to HIV testing and HIV positivity rates is shown in S1 Table. Our LCA explored models with up to six classes, with model fit statistics shown in Table 2. A four-class model was selected as the best fit for the data based on the BIC and sample-size-adjusted BIC values. Although models with more latent classes were associated with lower AIC/BIC values, the drop in BIC plateaued after five classes, and the LMR tests indicated no significant improvement in fit (p = 0.1969). The selected model had a sufficient entropy of 0.873, indicating good class separation.

Table 3 presents the conditional response probabilities, and the class counts of AYA in the four classes based on the most likely class assignment. We labelled each class based on the barriers with higher conditional probabilities as follows: "Low-risk perception", "Consent and

**Table 3. Conditional response probabilities of barriers to HIV testing, with latent class proportion for each class reported as a percentage next to class name.** Figures in bold show probabilities > 0.11, an arbitrary threshold selected to highlight the higher conditional response probabilities that informed the class labelling.; n is the final class count based on the most likely latent class.

| Barrier type | Low-risk perception (n = 7361, 39.50%) Probability | Consent and proximity (n = 5163, 27.74%) Probability | Testing site (n = 4996, 26.84%) Probability | Cost and logistics (n = 1092, 5.87%) Probability |
|---|---|---|---|---|
| Tests cost too much | 0.01 | 0.09 | 0.05 | **0.61** |
| Transport costs too much | 0.00 | 0.02 | 0.00 | **0.85** |
| Too far away | 0.02 | **0.13** | 0.03 | **0.57** |
| Don't know where to test | 0.02 | 0.03 | **1.00** | **0.41** |
| Cannot get treatment for HIV | 0.00 | 0.01 | 0.00 | 0.01 |
| Test kits not available | 0.01 | 0.05 | 0.01 | 0.03 |
| Afraid others will know about test result | 0.00 | 0.03 | 0.01 | 0.01 |
| Afraid spouse/partner/family will know results | 0.00 | 0.02 | 0.00 | 0.04 |
| Don't want to know I have HIV | 0.01 | 0.05 | 0.01 | 0.01 |
| Don't need tests/low-risk | **1.00** | 0.08 | 0.10 | 0.11 |
| Did not receive permission from spouse/family | 0.02 | **0.16** | 0.04 | 0.02 |
| Religious reasons | 0.00 | 0.04 | 0.00 | 0.00 |

Figures show estimated conditional response probability for reported types of barriers to HIV testing. Figures in bold show the higher probabilities by latent class selected to highlight differences

proximity", "Testing site", and "Cost and logistics". The class counts and proportions based on most-likely assignment are shown in Table 3. The class counts, and proportions based on distributed probabilities are available in S2 Table.

For the sensitivity analysis, our LCA explored models with up to seven classes, with model fit statistics shown in S2 File. Similar to result from imputed dataset, a four-class model was selected as the best fit for the data with similar class counts and distribution of AYA (33.41%, 28.63%, 24.26%, 13.70%). However, the conditional response probabilities of the barrier types differed from those obtained from the imputed dataset, as shown in S2 File.

## Sociodemographic and sex-behaviour characteristics of latent classes

The distribution of sexual activity and sociodemographic characteristics by latent classes is shown in S3 Table. As shown in Fig 3, the distribution of AYA across latent classes varied geographically. AYA in the low-risk perception class showed nearly equal distribution between the Northern and Southern zones, ranging from 14.4% to 17.1% in the Northern zones and from 14.5% to 19.5% in the Southern zones. The majority (67.1%) of AYA in the cost and

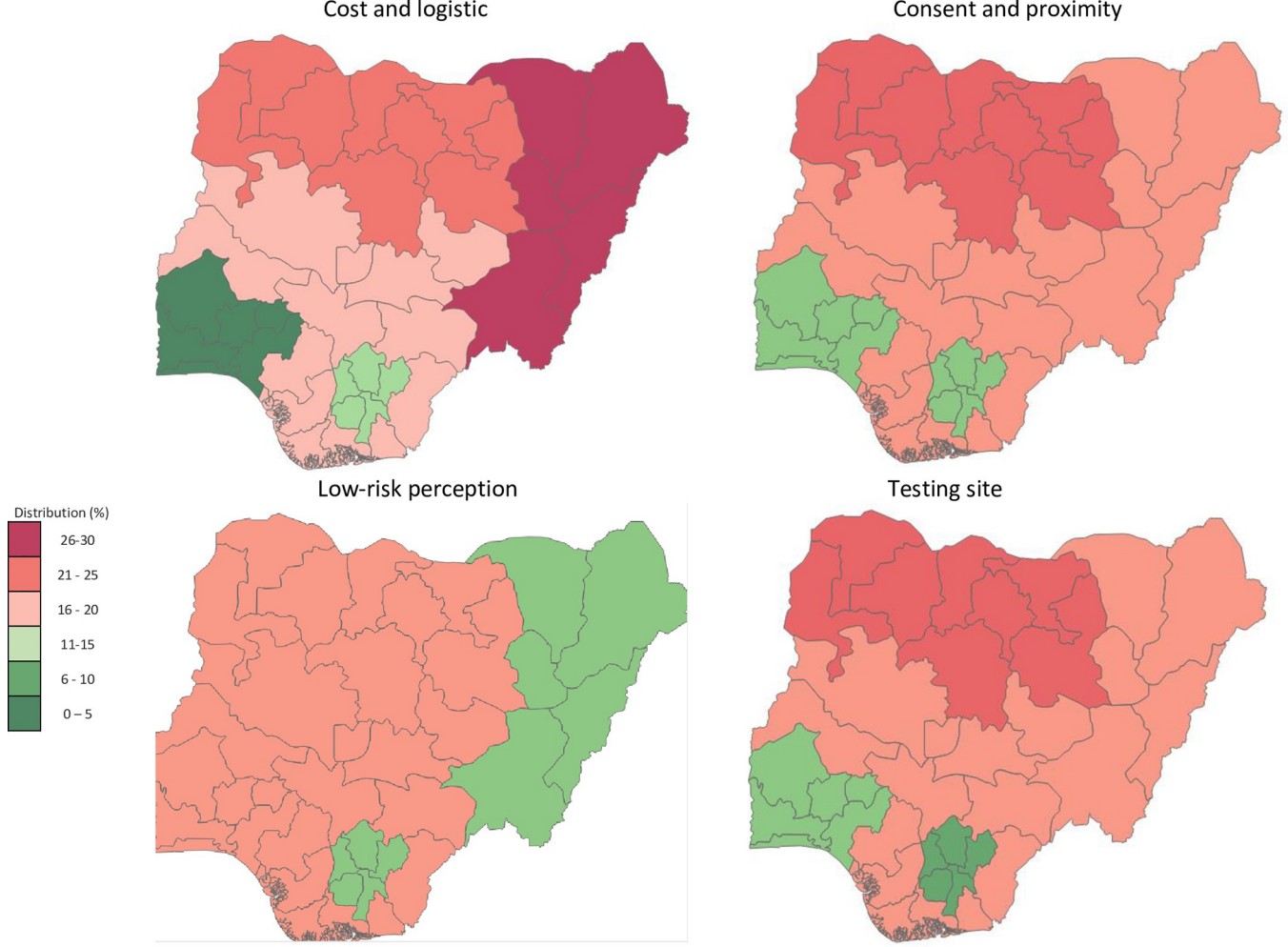

**Fig 3. Spatial map showing zonal geographical distribution of AYA by latent class subgroups.** Map data available from ©OpenStreetMap under the Open Database License.

logistics class resided in the Northern zones, ranging from 15.7% (Northcentral) to 28.4% (Northeast). Similarly, 60% of AYA in the testing site class reside in the Northern zones, ranging from 18.5% (Northeast) to 21% (Northwest). Additionally, 60% of AYA in the consent and proximity class reside in the Northern zones, ranging from 18.3% (Northeast) to 21.2% (Northwest).

Table 4 shows results of the LCR with sociodemographic factors and sexual activity, using the low-risk perception class as the reference category due to its status as the class with the highest proportion of AYA. Multicollinearity diagnostics revealed high positive correlation (0.917) between intergenerational sex and condom use at last sex with non-marital, non-cohabiting partners in the past 12 months. However, VIFs for both variables (6.34 and 7.45, respectively) were below the commonly accepted threshold of 10, indicating that multicollinearity was not severe enough to warrant variable removal [46]. Therefore, both variables were retained in the regression models. Correlation matrix and VIFs are presented in S2File.

Compared to AYA in the low-risk perception class, AYA in the consent and proximity class were more likely to reside in the Northern regions (aOR 1.80, 95% CI 1.35–2.41 for Northwest; aOR 1.79, 95% CI 1.33–2.42 for Northeast; aOR 1.98, 95% CI 1.52–2.56 for North central) and less likely to be married (aOR 0.71, 95% CI 0.56–0.92). They were more likely to engage in intergenerational sex with partners above 18 years (aOR 1.17, 95% CI 1.02–1.35), more likely to have two or more sexual partners (aOR1.75, 95% CI 1.39–2.20), and engaging in transactional sex (aOR1.50, 95% CI 1.23–1.84). They were less likely to use condoms at the last sex with non-marital, non-cohabiting partners (aOR 0.79, 95% CI 0.63–0.99).

AYA in the testing site class were more likely to reside in the Northern region (aOR 1.70, 95% CI 1.23–2.35 for Northwest; aOR 1.70, 95% CI 1.20–2.40 for Northeast; aOR 1.80, 95% CI 1.38–2.36 for North central) and have a rural residence (aOR 1.25, 95% CI 1.01–1.53). They were also less likely to be educated up to tertiary level (aOR 0.70, 95% CI 0.52–0.95) or be in the top three wealth quintiles (aOR 0.74, 95% CI 0.57–0.95). They were less likely to be married (aOR 0.76, 95% CI 0.59–0.97), more likely to engage in intergenerational relationships with partners above 18 years (aOR 1.21, 95% CI 1.04–1.39) and more likely to engage in transactional sex (aOR1.53, 95% CI 1.26–1.85).

AYA in the cost and logistic class were more likely to be aged 20–24 years (aOR 1.36, 95% CI 1.02–1.81), have rural residence (aOR 1.58, 95% CI 1.04–2.39), they were less likely to be females (aOR 0.74, 95% CI 0.56–0.97), less likely to have at least primary levels of education (aOR 0.63, 95% CI 0.42–0.94), and less likely to belong to middle or higher wealth quintiles (aOR 0.47, 95% CI 0.31–0.71). They were less likely to use condoms (aOR 0.58, 95% CI 0.34–0.98), know their partners' HIV status (aOR 0.40, 95% CI 0.25–0.63) and more likely to engage in transactional sex (aOR 2.12, 95% CI 1.58–2.84).

To determine the profile of AYA in the low-risk perception class, we reset the reference category by using each of the three other classes individually as reference categories in our analysis. Figures are shown in S4 Table. Compared to other categories, AYA in the low-risk perception class were more likely to reside in the South, to be females, aged 15–19 years, have urban residence, have an education, and belong to the top three wealth quintiles. They were less likely to engage in intergenerational sex, have two or more sexual partners, or engage in transactional sex. They were more likely to know their partner's HIV status and to use condoms.

## Latent class membership and HIV positivity rate

Fig 4 shows HIV positivity rates for the four classes with the national rates among AYA aged 15–24. [9] LCR with distal outcome showed that AYA in the consent and proximity class had a

**Table 4. Sociodemographic and sexual lifestyle correlates of latent classes.** The reference category for odds ratio (OR) is the Low-risk perception class. Bolded values indicate statistical significance as determined by 95% CIs not overlapping 1.00.

| | | Consent and proximity (n = 5163, 27.74%) | Testing site (n = 4996, 26.84%) | Cost and logistics (n = 1092, 5.87%) |
|---|---|---|---|---|
| Sociodemographic factors | | Adjusted OR (95% CI) | Adjusted OR (95% CI) | Adjusted OR (95% CI) |
| Age | 15–19 | 1.00 | 1.00 | 1.00 |
| | 20–24 | 1.11(0.94–1.32) | 0.92(0.78–1.08) | **1.36(1.02–1.81)** |
| Sex | Male | 1.00 | 1.00 | 1.00 |
| | Female | 1.08(0.94–1.23) | 0.97(0.85–1.12) | **0.74(0.56–0.97)** |
| Education | No education | 1.00 | 1.00 | 1.00 |
| | Primary | 0.97(0.76–1.22) | 0.91(0.73–1.13) | **0.63(0.42–0.94)** |
| | Secondary | 0.92(0.75–1.13) | 0.86(0.70–1.07) | 0.78(0.53–1.14) |
| | Tertiary | 1.13(0.85–1.50) | **0.70(0.52–0.95)** | 0.61(0.33–1.13) |
| | Other | 0.81(0.60–1.11) | **0.72(0.53–0.98)** | **0.38(0.22–0.66)** |
| Wealth quintile | Lowest | 1.00 | 1.00 | 1.00 |
| | Second | 0.89(0.71–1.12) | 0.96(0.77–1.19) | 0.91(0.67–1.25) |
| | Middle | 0.80(0.63–1.03) | **0.74(0.57–0.95)** | **0.47(0.31–0.71)** |
| | Fourth | 0.82(0.63–1.08) | 0.88(0.66–1.17) | **0.56(0.34–0.91)** |
| | Highest | 0.79(0.59–1.08) | **0.71(0.52–0.99)** | **0.19(0.09–0.40)** |
| Place of residence | Urban | 1.00 | 1.00 | 1.00 |
| | Rural | 0.94(0.77–1.16) | **1.25(1.01–1.53)** | **1.58(1.04–2.39)** |
| Religion | Christian | 1.00 | 1.00 | 1.00 |
| | Islam | **0.80(0.67–0.96)** | 0.85(0.7–1.02) | **0.53(0.33–0.88)** |
| | Other | 1.25(0.74–2.11) | 0.62(0.33–1.18) | 0.81(0.28–2.35) |
| Marital status | Never married | 1.00 | 1.00 | 1.00 |
| | Married/living with partner | **0.71(0.56–0.92)** | **0.76(0.59–0.97)** | 0.90 (0.58–1.41) |
| | Divorce/separated/widowed | 1.03(0.74–1.45) | 0.77(0.52–1.15) | 1.16(0.61–2.19) |
| Zone | Southwest | 1.00 | 1.00 | 1.00 |
| | Northwest | **1.80 (1.35–2.41)** | **1.70(1.23–2.35)** | **8.04(2.72–23.74)** |
| | Northeast | **1.79(1.33–2.42)** | **1.70(1.20–2.40)** | **7.87(2.92–21.19)** |
| | Northcentral | **1.98(1.52–2.56)** | **1.80(1.38–2.36)** | **4.75(1.81–12.47)** |
| | Southeast | 1.11(0.88–1.4) | 0.88(0.68–1.13) | **3.46(1.30–9.22)** |
| | Southsouth | 1.06(0.83–1.34) | 1.17(0.92–1.49) | **5.25(1.99–13.82)** |
| Sexual activity | | OR (95% CI) | OR (95% CI) | OR (95% CI) |
| Intergenerational sex | No | 1.00 | 1.00 | 1.00 |
| | Yes (with minor) | 1.16(0.84–1.59) | 1.19(0.87–1.63) | 1.18(0.6–2.32) |
| | Yes (with non-minor) | **1.17(1.02–1.35)** | **1.21(1.04–1.39)** | 1.14(0.88–1.48) |
| Condom use at last sex with non-marital, non-cohabiting partners in the past 12 months | No sex with non-marital, non-cohabiting partners in the past 12 months | 1.00 | 1.00 | 1.00 |
| | Condom not used | **0.79(0.63–0.99)** | 0.82(0.66–1.02) | **0.58(0.34–0.98)** |
| | Condom used | 0.82(0.66–1.02) | 1.03(0.85–1.25) | 0.78(0.48–1.26) |
| Number of sexual partners in the past 12 months | 0 | 1.00 | 1.00 | 1.00 |
| | 1 | 1.15(0.94–1.40) | 1.11(0.91–1.35) | 1.09(0.71–1.67) |
| | Two or more | **1.75(1.39–2.20)** | 1.24(0.98–1.57) | 1.31(0.79–2.15) |
| Partner HIV status known | No | 1.00 | 1.00 | 1.00 |
| | Yes | 0.93(0.79–1.1) | 1.04(0.87–1.24) | **0.40(0.25–0.63)** |

*(Continued)*

**Table 4.** (Continued)

|  |  | Consent and proximity (n = 5163, 27.74%) | Testing site (n = 4996, 26.84%) | Cost and logistics (n = 1092, 5.87%) |
|---|---|---|---|---|
| Transactional sex | No | 1.00 | 1.00 | 1.00 |
|  | Yes | **1.50(1.23–1.84)** | **1.53(1.26–1.85)** | **2.12(1.58–2.84)** |

All sociodemographic factors and sexual activity variables were mutually adjusted for one another. Please refer to Table 1 for a detailed description of each variable. Bold font highlights statistically significant difference (95% CI not crossing 1.00) from the reference group. Percentages and ORs allow for complex survey design features

higher likelihood of testing HIV positive (OR 1.68; 95% CI: 1.04–2.71). However, this association was no longer significant after adjusting for sociodemographic factors and sex behaviour (aOR 1.54, 95% CI 1.01–2.36). Adjusted analysis showed female sex, age group 20–24 years, being married or living with a partner, having 'other' types of education, of the Christian religion and residing in Southsouth zone as factors associated with a higher likelihood of testing HIV positive. The results of crude and adjusted analysis are shown in Table 5.

Adjusted analysis during sensitivity analysis showed significant association between barrier subgroup membership and HIV status, with condom use at last sex with non-marital, non-cohabiting partners in the past 12 months, two or more sexual partners in the past 12 months increasing the likelihood of being HIV infected while being female, knowing partners HIV status, and being aged 19–24 years reducing the likelihood of being HIV infected. See S2 File.

## Discussion

To our knowledge, this is the first study to explore patterns of barriers to HIV testing services among AYA in the context of sexual and sociodemographic characteristics and HIV status, using a nationally representative data. Among sexually active AYA in Nigeria who had never undergone HIV testing, we identified four subgroups based on the pattern of barriers reported in accessing HIV testing. The subgroups were termed low-risk perception (n = 7,361; 39.5%), consent and proximity (n = 5,163; 27.74%), testing site (n = 4,996; 26.84%), and cost and

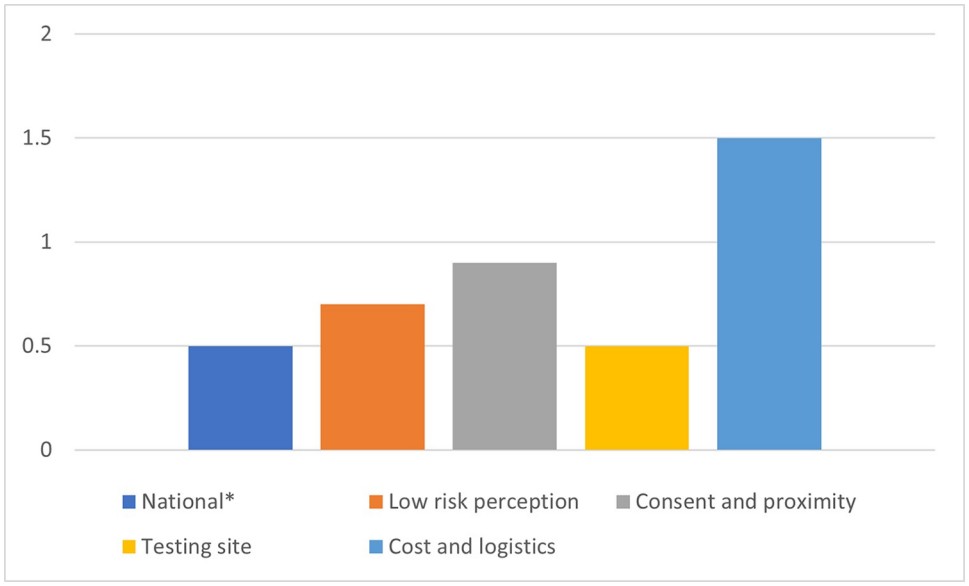

**Fig 4. HIV positivity rate in barrier classes.** *Source: NAIIS technical report.

**Table 5. Crude and adjusted association between latent class and HIV positivity rates.**

| Crude analysis | | OR (95%CI) |
|---|---|---|
| Latent class | Low-risk perception | 1.00 |
| | Consent and proximity | 1.68 (1.04–2.71) |
| | Testing site | 1.01 (0.57–1.77) |
| | Cost and logistics | **1.49 (0.65–3.39)** |
| **Adjusted analysis** | | Adjusted OR (95 CI) |
| Latent class | Low-risk perception | 1.00 |
| | Consent and proximity | 1.54 (1.01–2.36) |
| | Testing site | 1.02(0.63–1.66) |
| | Cost and logistics | 1.35(0.66–2.76) |
| Sociodemographic factors | | |
| Age | 15–19 | 1.00 |
| | 20–24 | **2.01(1.23–3.28)** |
| Sex | Male | 1.00 |
| | Female | **3.1(1.78–5.40)** |
| Education | No education | 1.00 |
| | Primary | 1.73(0.91–3.28) |
| | Secondary | 0.75(0.33–1.74) |
| | Tertiary | 0.85(0.27–2.64) |
| | Other | **2.69(1.04–6.95)** |
| Wealth quintile | Lowest | 1.00 |
| | Second | 0.56(0.29–1.09) |
| | Middle | 1.02(0.47–2.17) |
| | Fourth | 1.26(0.55–2.88) |
| | Highest | 0.88(0.32–2.36) |
| Place of residence | Urban | 1.00 |
| | Rural | 1.08(0.67–1.76) |
| Religion | Christian | 1.00 |
| | Islam | **0.24(0.11–0.52)** |
| | *Other | 0.00 (0.00–0.00) |
| Marital status | Never married | 1.00 |
| | Married/living with partner | **6.81(2.01–23.05)** |
| | Divorce/separated/widowed | 0.05(0–0.84) |

(*Continued*)

**Table 5.** (Continued)

| Crude analysis | | OR (95%CI) |
|---|---|---|
| Zone | Southwest | 1.00 |
| | Northwest | 0.52(0.17–1.55) |
| | Northeast | 2.09(0.86–5.07) |
| | Northcentral | 1.52(0.71–3.24) |
| | Southeast | 1.34(0.62–2.90) |
| | Southsouth | **2.40(1.25–4.60)** |
| Sexual behaviour | | |
| Intergenerational sex | No | 1.00 |
| | Yes (with minor) | 2.0 (0.73–5.46) |
| | Yes (with non-minor) | 1.44(0.93–2.23) |
| Condom use at last sex with non-marital, non-cohabitating partners in the past 12 months | No sex with non-marital, non-cohabitating partners in the past 12 months | 1.00 |
| | Sex, condom not used | 1.65(0.81–3.37) |
| | Sex, condom used | 1.94(0.91–4.11) |
| Number of sexual partners in the past 12 months | 0 | 1.00 |
| | 1 | 0.80 (0.45–1.44) |
| | Two or more | 1.08 (0.52–2.25) |
| Partner HIV status known | No | 1.00 |
| | Yes | 0.62 (0.19–2.01) |
| Transactional sex | No | 1.00 |
| | Yes | 0.89 (0.53–1.49) |

All sociodemographic factors and sexual activity variables were mutually adjusted for one another. Please refer to Table 1 for a detailed description of each variable. Figures in bold represent statistical significance as determined by 95% CIs not overlapping 1.00.

*Other refer to traditional religion, no religion, and other types of religion.

logistics (n = 1,092; 5.87%). Although previous research in Nigeria has identified all the twelve individual barriers reported in our study [18, 37, 38], our study makes important advancement by delving deeper into how these barriers group together within specific segments of the AYA population, which might inform more tailored combination interventions aiming to simultaneously address multiple needs. For instance, while distance presents a common obstacle in the "Consent and proximity" and "cost and logistics" categories, their unique combinations with other barriers call for different interventions. In the "Consent and proximity" category, the coexistence of testing permission denial and distance barriers could indicate autonomy as the main problem, as autonomy limitation could reflect mobility restriction and lack of independence in accessing testing services. Conversely, the combination of distance, knowledge, and cost barriers within the "cost and logistics" category points to predominantly access

challenges with geographical and economic gaps. These challenges require distinct interventions compared to those needed for AYA in the "Consent and proximity" subgroup. Thus, using LCA to situate each barrier in its specific context enables a more comprehensive understanding that can inform the development of well-tailored and potentially more effective combination interventions [33].

We found that the AYA subgroups had shared and unique sexual behaviour characteristics. While AYA in the "low-risk perception" class exhibited sexual behaviours consistent with their perception of having low likelihood of HIV acquisition, the slightly elevated HIV positivity rates in this group suggest potential underestimation of this risk, consistent with previous studies show high rates of inaccurate risk perception among AYA [19, 33, 47–50]. Such inaccuracies represent a significant concern for national HIV prevention efforts, given that significant proportions of AYA in the low-risk perception are distributed across four of six zones.

Our observation of AYA in the consent and proximity class suggests that limitation in autonomy could be a common driver of non-utilisation of HIV testing services and observed sex behaviour [34, 51]. Specifically, intergenerational and transactional sex, and multiple concurrent sexual relationships shown in this group have all been linked with limited autonomy within the family or marital context with relationship power imbalances [16, 52, 53]. The prominence of this subgroup within the Northwest region could be explained by well-documented cultural limitations in the sexual and reproductive health rights (SRHR) for women and young people in northern Nigeria [54–56], with often restricted autonomy in healthcare decisions and access [54–56]. Studies in Nigeria and elsewhere [54–56] have linked limited autonomy with poor reproductive, maternal and child health outcomes. Given the ongoing shift towards biomedical approaches to HIV prevention and treatment, such as pre-exposure prophylaxis (PreP) and treatment as prevention (TasP), our study emphasises the need to integrate autonomy-building interventions into health programs as well as address broader sociocultural barriers that limit autonomy and SRHR of women and young people [54, 56–63]. Similarly, increased likelihood of intergenerational sex, and transactional sex among AYA in the "testing site" subgroup suggests that informational gaps in HIV prevention information and intervention, including where and how to access testing services, could be a common driver of non-utilisation of HIV testing services and sexual activity observed in this subgroup. The prominence of this subgroup in the Northwest is consistent with studies [64, 65] in Nigeria that have shown that while newer and convenient HIV testing methods such as home-based and self-testing are being introduced to address access barriers to HIV testing, awareness and acceptance remain low in Northern Nigeria. This suggests the need for comprehensive strategies involving education, awareness campaigns, improved accessibility, and stigma reduction efforts.

The proportion of AYA in the "cost and logistics" subgroup is notably higher in the Northeast region, likely linked to the historically poor socioeconomic development of the region, compounded by prolonged conflicts, which might also explain the observed association of the group with rural residence, belonging to the lowest wealth quintiles, and lower educational attainment [40, 66]. Given the subgroup's elevated proportion of AYA with HIV positive status, the increased likelihood of condomless sex, transactional sex, and lower likelihood of knowledge of partners' HIV status emphasises the central role of sociodemographic and structural factors in shaping overall HIV vulnerability, including impacting health behaviour and barriers to access [17, 34, 67–70]. Notably, our adjusted analysis showed that sociodemographic factors, rather than barrier types, better explain variation in HIV status. In line with other studies, we found that being female [5], age-group 20–24 years [5, 34], being married or living with a partner elevated the risk of HIV infection [13, 54]. In addition, the elevated

likelihood of testing HIV positive in the Southsouth zone is consistent with regional HIV positivity trend and pattern in the country [70].

The combination of HIV positivity rates, sociodemographic factors, and sexual behaviour profiles within the four identified classes offers valuable insights into the potential trajectories of HIV vulnerability that AYA might follow. Understanding these trajectories creates an opportunity for early identification and intervention. Notably, although AYA in the "consent and proximity" and "low-risk perception" classes share similar sociodemographic characteristics, they exhibit markedly different sexual behaviours and HIV infection odds, suggesting that they may represent divergent vulnerability patterns within the same AYA subgroup. Implementing interventions that promote autonomy among young people at an early stage of their development can create a favourable environment for positive transitions from the "consent and proximity" profile to the "low-risk perception" profile [58, 59]. This transition increases the likelihood of adopting the "low-risk perception" profile, characterised by sexual activity that reduces vulnerability to HIV. While we acknowledge that the "low-risk perception" profile also presents unique challenges requiring targeted interventions, this proactive approach can lead to long-term benefits in HIV prevention.

Our study has important strengths. Firstly, we utilised a nationally representative dataset with a well-structured design, increasing our findings' generalizability. Furthermore, using LCR, we identified AYA subgroups with shared and distinctive patterns of HIV testing barriers, sexual activity, sociodemographic characteristics, and HIV positivity rates. Our approach and findings pave the way for the development of tailored, person-centered, and youth-friendly services that cater to different AYA subgroups' specific needs and preferences [35].

Our study also comes with limitations that offer opportunities for further research. Firstly, the cross-sectional design of the NAIIS restricts our ability to establish causal relationships [71]. Secondly, using self-reported data in the NAIIS may introduce recall and social desirability biases, potentially biasing our estimates [9]. Additionally, the absence of separate analyses for males and females might mask sex-specific differences in latent class composition and other factors we investigated, especially considering the varying reliability of self-reported data between males and females concerning sexual behaviour [72]. Although we attempted to mitigate this limitation by adjusting for sex and other factors when examining the relationship between latent classes and HIV infection, some potential inaccuracies may persist. Moreover, the use of only vaginal sex as a measure of sexual activity in NAIIS excludes the experiences and activities of sexual minority groups, for example. In addition to this, the binary coding of sex in NAIIS excludes transgender and other gender expansive individuals from our analysis. These are significant communities in the context of the HIV epidemic in Nigeria and other settings [73, 74] and future research should focus on sexual behaviours and the current HIV epidemic in these communities. Also focusing solely on vaginal sex and sexual activity to explain HIV positivity might have led us to overlook the significant contribution of other non-sexual transmission routes, such as injection drug use. This oversight is crucial, especially considering that injection drug use is increasingly becoming a significant driver of the HIV epidemic among young people in Nigeria [70]. Additionally, the recoding of variables like religion may obscure important subgroup identities [9]. Lastly, using Latent Class Analysis (LCA) carries the potential for classification errors when determining the most likely class assignment, despite our efforts to minimise this using the integrated 3-step approach in Mplus [27, 43].

The relative consistency in identifying an optimal four-class LCA model and the similarity in class distribution during sensitivity analysis strengthens the reliability of our findings. However, discrepancies between findings from imputed and non-imputed datasets, particularly in the association between barrier subgroup membership and HIV status, and variations in sociodemographic and sexual factors linked to HIV infection, necessitate careful interpretation. As

we did not verify whether missing data were missing at random (MAR) or missing completely at random (MCAR), our application of multiple imputation, considered a best practice for handling missing data, may potentially introduce biases in the imputed data [75]. Thus, our results should be interpreted with caution, taking cognisance of findings from both imputed and non-imputed analyses. Nonetheless, the findings present an opportunity for further research. Exploring alternative data handling methods, including multiple imputation with more sophisticated models, can strengthen the robustness of our findings and deepen our understanding the role of barriers to HIV testing, sexual and sociodemographic factors in shaping vulnerability to HIV in adolescents and young people.

## Conclusion

In conclusion, our study identified distinct subgroups of AYA based on patterns of barriers to HIV testing services and assessed the association between these barrier patterns and sexual behaviour, socio-demographics, and HIV status between AYA subgroups. We showed that patterns of barriers to HIV testing are linked with differences in sexual behaviour and sociodemographic profiles among AYA, with the latter driving differences in HIV positivity rates. Findings can improve combination healthcare packages aimed at simultaneously addressing multiple barriers and determinants of vulnerability to HIV among AYA. This is highly pertinent as countries face challenges in striking the right balance of targeted and comprehensive programming for AYA.

## Supporting information

**S1 File. National algorithm for HIV testing used in NAIIS 2018.**
(TIF)

**S2 File. Sensitivity analyses and correlation diagnostics.**
(TIF)

**S1 Table. Characteristics of study sample.**
(TIF)

**S2 Table. Final class counts and proportions for latent classes based on distributed probabilities.**
(TIF)

**S3 Table. Sociodemographic and sex behaviour correlates of latent classes.**
(TIF)

**S4 Table. Latent class comparison using alternate reference classes.**
(TIF)

## Acknowledgments

We thank the Nigeria Federal Ministry of Health for granting study data access and approval. The authors also thank the adolescents and young adults who consented to using their data for research.

## Author Contributions

**Conceptualization:** Okikiolu Badejo, Edwin Wouters, Anne Buve, Tom Smekens, Plang Jwanle, Christiana Nöstlinger.

**Data curation:** Okikiolu Badejo, Edwin Wouters, Tom Smekens, Christiana Nöstlinger.

**Formal analysis:** Okikiolu Badejo, Edwin Wouters, Anne Buve, Tom Smekens, Christiana Nöstlinger.

**Funding acquisition:** Okikiolu Badejo, Edwin Wouters, Sara Van Belle, Marie Laga, Christiana Nöstlinger.

**Investigation:** Okikiolu Badejo, Sara Van Belle.

**Methodology:** Okikiolu Badejo, Tom Smekens.

**Project administration:** Edwin Wouters, Plang Jwanle, Marie Laga, Christiana Nöstlinger.

**Supervision:** Sara Van Belle, Anne Buve, Christiana Nöstlinger.

**Validation:** Sara Van Belle, Christiana Nöstlinger.

**Writing – original draft:** Okikiolu Badejo.

**Writing – review & editing:** Okikiolu Badejo, Edwin Wouters, Sara Van Belle, Anne Buve, Tom Smekens, Plang Jwanle, Marie Laga, Christiana Nöstlinger.

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
