## [Decision Letter · Decision Letter 0]

27 Sep 2023

PONE-D-23-19074Intersecting HIV testing barriers, sexual behaviour and undiagnosed HIV: a latent class approach to understanding HIV vulnerability among adolescents and young adults in NigeriaPLOS ONE

Dear Dr. Badejo,

Thank you for submitting your manuscript to PLOS ONE. After careful consideration, we feel that it has merit but does not fully meet PLOS ONE’s publication criteria as it currently stands. Therefore, we invite you to submit a revised version of the manuscript that addresses the points raised during the review process.

We look forward to receiving your revised manuscript.

Kind regards,

Belayneh Mengist, MPH

Academic Editor

PLOS ONE

“OB was supported with research funding from the Belgium Directorate-General for Development Cooperation (DGD) awarded through the Institute of Tropical Medicine Antwerp, Belgium. SVB received funding from the Flanders Research Foundation (FWO) grant number 1221821N.”

Please include this amended Role of Funder statement in your cover letter; we will change the online submission form on your behalf."

4. We note that Figure 2 in your submission contain [map/satellite] images which may be copyrighted. All PLOS content is published under the Creative Commons Attribution License (CC BY 4.0), which means that the manuscript, images, and Supporting Information files will be freely available online, and any third party is permitted to access, download, copy, distribute, and use these materials in any way, even commercially, with proper attribution. For these reasons, we cannot publish previously copyrighted maps or satellite images created using proprietary data, such as Google software (Google Maps, Street View, and Earth). For more information, see our copyright guidelines: http://journals.plos.org/plosone/s/licenses-and-copyright.

1. You may seek permission from the original copyright holder of Figure 2 to publish the content specifically under the CC BY 4.0 license. 

Reviewers' comments:

Reviewer's Responses to Questions

**Comments to the Author**

1. Is the manuscript technically sound, and do the data support the conclusions?

Reviewer #1: Yes

Reviewer #2: Yes

2. Has the statistical analysis been performed appropriately and rigorously? 

Reviewer #1: Yes

Reviewer #2: Yes

3. Have the authors made all data underlying the findings in their manuscript fully available?

Reviewer #1: Yes

Reviewer #2: No

4. Is the manuscript presented in an intelligible fashion and written in standard English?

Reviewer #1: Yes

Reviewer #2: No

5. Review Comments to the Author

Reviewer #1: Abstract

You have mentioned how less is known about barriers interacting across multiple levels, but have not cited the applicability of LCA to be able to capture this. Would be useful to comment on LCA’s ability to derive homogeneity between different barriers and thus capture their intersection, as well as maybe a brief statement about what LCA is (a mixture-modelling technique deriving a latent variable etc.).

Line 43: Is sexual behaviour an actual covariate or is it a theme of covariates? Sounds very broad, would be useful to list the actual covariates used. Needs a full stop at the end of the paragraph.

Lines 44-45: Please clarify in the abstract that the bracketed values are the proportion of the population that are assigned to each latent class, might not be clear.

Lines 46-48: Again, would be useful to get an indication of what these sexual practices are that you are measuring when you say riskier sexual behaviours.

Lines 48-50: Consistency in reporting of HIV prevalence, you have reported proportions for all except test-permission denied, which you have reported an OR.

Line 51: What sociodemographic factors? Also, confusing about which classes have higher odds of HIV infection, is it just low-risk perception which you have reported the OR for, or is it also test-permission denied and test-permission denied.

Line 52: Full stop at end of paragraph.

Introduction:

Line 69: Citation for end to AIDS pandemic.

Line 82: need citation for period of adolescence and use being characterised by increased sexual risk taking, unless it is the same for the point about risk taking being associated with increased HIV acquisition?

Line 85: citation needed

Line 92: citation needed

Line 96: citation needed, even If the same as that in line 99

Line 97: maybe instead of “according to the UNAIDS 95-95-95 cascade” you could say “Appraising Nigeria’s current HIV landscape against the UNAIDS 95-95-95 cascae”

Introduction would benefit from a methodological introduction to LCA, and how this technique can be used to “account for the heterogeneity within the AYA population” and the “complex relationships between factors at different levels”

Line 107: citation needed

Methods:

Line 125: citation needed

Line 126: citation needed

Lines 138-140: Would be useful to know the response rates of the surveys

Figure 1: So vaginal sex was used to determined whether people were sexually active? That is an important thing to note in the methods, as it is making an important and perhaps controversial, definition of sexually active. For example, it is likely to be excluding sexual minority groups who do not enact this type of sex, or those who practice primarily non-penetrative sex. Understand it is probably too late to change this now, however this has to be noted as the definition of sexually active in line 159, and has to be highlighted as a limitation in the discussion, particularly mentioning its potential exclusion of sexual minority groups (who are a key population within the HIV epidemic of Nigeria - https://bmcpublichealth.biomedcentral.com/articles/10.1186/s12889-019-7540-4 ). Also negates the potential transmission of HIV through oral sex, which is another limitation that should be mentioned.

Lines 181-185: Don’t think you need this given that you have already mentioned the variables you have been investigating.

Line 194: final HIV status from the blood test?

Table 1: fine to recode religion into a binary category as this is often required to perform regression analyses on an LCA model, but the limitations of this in being reductive and potentially missing important religious subgroups/denominations should be mentioned in the discussion.

Table 1: Be consistent in the table with putting ‘Not recoded’, as it is missing from ‘Gender’, for example. Would be interested to see if there were any trans/non-binary patients who did not answer this question, and how their data was handled. If individuals not answering the gender question were excluded, this is a very important limitation to cite.

Table 1: Martial status recode column is the same as that for religion.

Table 1: Gender question will not capture those who are trans or non-binary. Similarly, if the data does not present this there is nothing you can do, however, it is important to mention this in the discussion, with this again being an important population in the Nigerian HIV epidemic (https://www.ncbi.nlm.nih.gov/pmc/articles/PMC7906338/)

Line 198: cite STATA here.

Line 203: cite Mplus here.

Line 212: You need to state clearly which variables were used as manifest variables to construct the classes, and how these were chosen (i.e., previous research, previous LCA models)

Line 209: citation needed.

Line 210: missing a closed bracket at the end of the sentence

Line 214-215: don’t think you need the expansion of the maximum probability rule, think the space would be better use explaining what the LCA model is

Line 230: cite this

Lines 240-254: think this should go in the Ethics statement at the beginning of the paper.

Results:

Understand if space is limited, but would be interesting to get an idea of the proportions of the sample who reported each barrier to HIV testing (those highlighted in lines 170-177)

Line 257: reference figure 1 here, and potentially consider moving this figure to this point in the paper.

Line 257: ‘Sample characteristics’. When you say overall study, does this mean that this is the demographics of the 18,612 participants in the LCA model, or the overall 229,762 participants in NAIIS? If it is the latter, I think this table should be redone to show the characteristics of only those who are in the LCA model.

Line 268: cite this national estimate figure

Table 2: “Yes (with non-minor” needs a closing bracket.

Table 2: capitalise ‘missing’ throughout’

Table 2: incomplete, need to fill in the final HIV status

Table 3: No need to report after 5 classes, up to here is sufficient to show the drop in AIC/BIC which plateaus after 4 class model, and the rising LMR after 4 classes.

Line 282: replace probabilities with “conditional response probabilities”, and this term should be explained in the methods.

Table 5: If this the characteristics of the sample in the LCA, then I would remove table 2 and keep this instead. Table 2 is otherwise misleading suggesting this is the demographics of those in the LCA.

Lines 298-311: include p values for those that are significant

Table 5: need to signify what the bolding means

Table 5: put �2 instead of ‘chi2’ in the table

Line 331: I think instead of association between classes, it should be ‘odds of class assignment’

Line 333-337: In methods section, there needs to be an indication that statistical significance will be assessed by 95% confidence intervals of ORs not overlapping 1

Line 339: remove second comma

Line 343-344: rephrase this, doesn’t read well.

Line 344-345: ‘lower likelihood of knowing partners HIV status’

Table 6: can remove the low risk perception column as this looks messy having a blank table, and put that odds ratios are calculated relative to the low risk perception class in the title of the table.

Table 7 and figure 3 are overlapping on my copy of the manuscript, making Table 7 difficult to read.

It is a significant finding that the association between class membership and HIV infection disappeared, so I think that appendix table 3 should be integrated into table 7

Discussion

Line 382-384: It is important to acknowledge that this association disappeared when accounting for sociodemographic characteristics and behaviour

Line 406: LCA acronym can be introduced earlier

A theme that is not addressed sufficiently in the discussion is that the association of class membership and HIV prevalence disappears when accounting for sociodemographic and risk behaviours, and I think this should be commented upon, as this undermines the importance of barriers in HIV prevalence and implies the importance of sociodemographic characteristics (potentially indicating the role of structural prejudice in informing access) and sexual risk behaviours in driving increased HIV prevalence.

Line 466-467: not cited. In general I would recommend going back through the discussion and identifying where there unsupported statements that require some form of citation to support them.

Line 482: citation needed

Additional limitations that need to be mentioned: equating of vaginal sex with sexual activity and its subsequent impact on excluding key populations/excluding some potentially risky practices as mentioned above, no analysis on trans/non-binary individuals (even if data did not allow this, the exclusion of this population and its potential impact should be noted as this is a vulnerable population in Nigerian HIV epidemic), methodological limitations of LCA (see other LCA papers in the literature to get an indication of what these are), limitation of most-likely class assignment in not capturing those individuals for whom class assignment was ambiguous (i.e., someone who had a 49% likelihood of being assigned to “logistic constraint” and a 51% likelihood of being assigned to “low-risk perception” is not the same as someone who had a 100% likelihood of being assigned to “low-risk perception”).

General comments:

Good manuscript, but needs more citation throughout, more thorough description of the methodology of LCA, its limitations and why it is appropriate for this setting, and the potential key populations that are overlooked in this analysis (i.e., sexual and gender minorities).

Reviewer #2: Intersecting HIV testing barriers, sexual behaviour and undiagnosed HIV: a latent class

approach to understanding HIV vulnerability among adolescents and young adults in

Nigeria

This paper addresses a pertinent issue by examining the obstacles to HIV testing among a group that has received limited attention in research—adolescents and young individuals. By employing latent class analysis, multinomial regression, and binary logistic regression, the study offers significant findings regarding the correlates of HIV testing barriers and HIV infection. Nevertheless, the paper is not reader-friendly and major revisions are necessary for this paper prior to publications.

General comments

1. This paper contains numerous typos and grammatical errors that require correction before publication.

Specific comments

Title

2. The paper's title does not accurately represent its content. While this study does not primarily focus on the intersectionality of barriers to HIV testing, its main objective is to determine the factors or correlates associated with these barriers. To achieve this, the researchers employed LCA to condense the 12-item questionnaire on HIV testing barriers into four latent classes. These classes were then combined into a variable that served as the primary outcome, although this was not explicitly stated. It is important to note that the utilization of LCA in this study aimed to identify four distinct groups of individuals with similar response patterns to the questions regarding HIV testing barriers. The correlates of HIV were not assessed by LCA as stated in the title but with binary logistic regression. Furthermore, no analysis was conducted to demonstrate the co-occurrence of HIV testing barriers, sexual behavior, and undiagnosed HIV.

3. There is nothing in the paper that indicated “undiagnosed HIV”. This is a complex concept to measure.

Abstract

4. Page 1, lines 35-39: The introduction was incongruous with the rest of the paper.

5. Page 1, lines 37-39” “….and how these intersections impact patterns of sexual behavior and the prevalence of undiagnosed HIV among adolescents and young people in Nigeria.”: The study did not provide any specific evidence on how the intersections influenced sexual behavior or the prevalence of undiagnosed HIV among adolescents and young people in Nigeria. Additionally, factors related to sexual behavior were considered as potential confounding variables in the association between the latent classes of HIV testing barriers (primary outcome variable) and sociodemographic variables (independent factors). The objective of the study is unclear and needs to be rephrased.

6. Page 1, lines 40-43: The authors should clearly specify the methods employed to tackle both the primary and secondary objectives, as well as define the primary and secondary outcome variables. Based on the main findings, it seems that the authors utilized latent clusters as the primary outcome measure, and HIV status as the secondary outcome variable.

7. Page 1, line 43: The term "HIV prevalence" is misleading in this context. It appears that the authors were actually referring to the HIV status determined from the blood samples collected.

8. Page 1, line 44: I would suggest renaming the clusters to improve clarity. Here are my suggestions: Cluster 1: Low-risk perception cluster, cluster 2: Consent and proximity cluster, cluster 3: Testing site cluster and cluster 4: Cost and logistics cluster.

9. Page 1, line 46: Regarding HIV, it is important to clarify that the percentage of positive test results represents the HIV positivity rate, rather than prevalence. The term 'prevalence' encompasses a broader scope of HIV infection within a particular population." Recall that the study was conducted among AYA who had never been tested for HIV in Nigeria, so prevalence is not applicable.

10. Page 1, line 53: The conclusion does not correspond with the findings.

Introduction:

11. Page 2: The introduction is notably deficient in summarizing the existing body of literature regarding the identifiable factors that impact desired outcomes, particularly in relation to the barriers affecting HIV testing and HIV susceptibility among adolescents and young individuals.

12. Page 3, line 116: The second question asked about the influence of barriers to HIV testing on sociodemographic characteristics and sexual behaviors. However, the intended question should have been about the impact of sociodemographic characteristics and sexual behaviors on different clusters of barriers to HIV testing. Here is the revised question:

"What sociodemographic and sexual behaviors influence different clusters of barriers to HIV testing?"

13. Page 3, line 118: The third question appears to be unusual. It assessed the relationship between the clusters of barriers to testing and “undiagnosed HIV”. How can one determine undiagnosed HIV? In the methods section, it was mentioned that the participants had never undergone HIV testing. This suggests that individuals who tested positive contributed to the overall HIV positivity rate. Since the analysis is conducted at the individual level, it would be more appropriate to refer the outcome as HIV-positive.

Methods

14. Page 3, line 124: “Despite economic growth in recent decades, the country continues to face high poverty rates.” What is the method used to measure the poverty rate? Please provide a reference for the regional poverty rates.

15. Page 3, line 154: “According to the national algorithm”: The national algorithm should be stated.

16. Page 4, lines 164-194: It is challenging to determine which variables served as the independent and dependent variables. In contrast to what was stated as the outcome measure in line 194, the primary outcome measure was the barrier to HIV testing, while the secondary outcome measure was HIV status. It is important to state that barrier to HIV testing was selected as an independent variable for the secondary outcome analysis.

17. There is redundancy from lines 170-194 because the same information was presented in Table 1.

18. Table 1 should be summarized for clarity’s sake.

19. Table 1: The authors should note that gender and sex are not the same. While sex is a biological construct (male and female), gender is a social construct. Based on the question asked, sex is more appropriate.

20. Page 6, lines 202-238: Describe the statistical approaches in relation to the objectives. In addition, specify upfront that LCA was employed to reduce data dimensionality of barriers to HIV testing, as its mention arrived quite late in line 220. Also, what type of regression model was performed for HIV status?

Results

21. Table 2 is not clear. Which estimates are frequencies (small n) and medians?

22. How is final HIV status in Table 2 operationalized? Also, why is final HIV status placed under sexual activity?

23. Page 10, line 315 (Table 5): The value of p=0.000 does not exist. When the p-value is extremely low, statistical software approximates it to zero. In such cases, it is more appropriate to report the p-value as p<0.001.

24. Table 5: What is the reason behind certain cells being displayed in bold font?

25. Page 12, line 331: Multinomial regression can not determine association between latent classes. It is used to determine association between independent variables and outcome variable with more than two categories.

26. Page 13: Table 6: Please indicate what the reported magnitude of effect represents. Are they relative risk ratios (RRR) with 95%CI?

27. Page 14, line 373: “(OR 1.68; CI: 1.04-2.71, p =0.04)”: State that this is 95%CI.

28. Page 14, line 374: “The association between latent classes and HIV prevalence disappeared”. As stated previously, HIV positivity is more appropriate.

29. Table 7 should reflect crude or unadjusted model. The adjusted model is more important to the reader than Table 7 (unadjusted model).

Discussion

30. The identified factors influencing the barriers to HIV testing and HIV infection were not substantially discussed.

31. The spatial pattern of clusters of barriers to HIV testing was completed omitted in the discussion section.

32. Page 16, line 447: “The complete profiles (HIV prevalence, socio-demographic factors, and sexual behavior) of the four classes, taken together, offer an understanding of potential pathways through which AYA can transition along the spectrum of sexual risk and HIV vulnerability”: This statement is not accurate because pathways can be identified using LCA. If the authors are interested in identifying pathways, they need to run mediation or structural equation models.

6. PLOS authors have the option to publish the peer review history of their article (what does this mean?). If published, this will include your full peer review and any attached files.

Reviewer #1: **Yes: **Luke Muschialli

Reviewer #2: **Yes: **Daniel A. Adeyinka

---

## [Author Response · Author response to Decision Letter 0]

30 Oct 2023

Dear Dr. Belayneh Mengist,

Editor,

PLOS ONE

15 October 2023 

Resubmission of manuscript “Latent class analysis of barriers to HIV testing services and associations with sexual behaviour and HIV status among adolescents and young adults in Nigeria”

Dear Dr Mengist, 

We are pleased to submit our revised manuscript entitled: “Latent class analysis of barriers to HIV testing services and associations with sexual behaviour and HIV status among adolescents and young adults in Nigeria” to the PLOS ONE for consideration for publication. We appreciate the careful review and constructive suggestions. 

We believe that the manuscript is substantially improved after addressing editor and reviewers’ comments. Submitted with this letter are the editor and reviewers’ comments with our responses in tabular form, including how and where the text was modified. 

The revision has been developed in consultation with all co-authors, and each author has seen and approved the final form of this revision. The funders had no role in study design, data collection and analysis, decision to publish, or preparation of the manuscript.

Thank you for your consideration of our revised paper.

Sincerely, 

Okikiolu Badejo for the co-authors

---

## [Decision Letter · Decision Letter 1]

21 Nov 2023

PONE-D-23-19074R1Latent class analysis of barriers to HIV testing services and associations with sexual behaviour and HIV status among adolescents and young adults in NigeriaPLOS ONE

Dear Dr. Badejo,

Thank you for submitting your manuscript to PLOS ONE. After careful consideration, we feel that it has merit but does not fully meet PLOS ONE’s publication criteria as it currently stands. Therefore, we invite you to submit a revised version of the manuscript that addresses the points raised during the review process.

If applicable, we recommend that you deposit your laboratory protocols on protocols.io to enhance the reproducibility of your results. Protocols.io assigns your protocol its own identifier (DOI) so that it can be cited independently in the future. For instructions, see: https://journals.plos.org/plosone/s/submission-guidelines#loc-laboratory-protocols. Additionally, PLOS ONE offers an option for publishing peer-reviewed Lab Protocol articles, which describe protocols hosted on protocols.io. Read more information on sharing protocols at https://plos.org/protocols?utm_medium=editorial-email&utm_source=authorletters&utm_campaign=protocols.

We look forward to receiving your revised manuscript.

Kind regards,

Belayneh Mengist, MPH

Academic Editor

PLOS ONE

[Please do not edit.]

Reviewers' comments:

Reviewer's Responses to Questions

**Comments to the Author**

1. If the authors have adequately addressed your comments raised in a previous round of review and you feel that this manuscript is now acceptable for publication, you may indicate that here to bypass the “Comments to the Author” section, enter your conflict of interest statement in the “Confidential to Editor” section, and submit your "Accept" recommendation.

Reviewer #1: (No Response)

Reviewer #2: (No Response)

2. Is the manuscript technically sound, and do the data support the conclusions?

The manuscript must describe a technically sound piece of scientific research with data that supports the conclusions. Experiments must have been conducted rigorously with appropriate controls, replication, and sample sizes. The conclusions must be drawn appropriately based on the data presented.

Reviewer #1: Yes

Reviewer #2: Yes

3. Has the statistical analysis been performed appropriately and rigorously?

Reviewer #1: Yes

Reviewer #2: Yes

4. Have the authors made all the data underlying the findings in their manuscript fully available?

The PLOS Data policy requires authors to make all data underlying the findings described in their manuscript fully available without restriction, with a rare exception (please refer to the Data Availability Statement in the manuscript PDF file). The data should be provided as part of the manuscript or its supporting information or deposited in a public repository. For example, in addition to summary statistics, the data points behind means, medians, and variance measures should be available. If there are restrictions on publicly sharing data—e.g., participant privacy or use of data from a third party—those must be specified.

Reviewer #1: Yes

Reviewer #2: Yes

5. Is the manuscript presented in an intelligible fashion and written in standard English?

Reviewer #1: Yes

Reviewer #2: Yes

6. Review Comments to the Author

Please use the space provided to explain your answers to the questions above. You may also include additional comments for the author, including concerns about dual publication, research ethics, or publication ethics. (Please upload your review as an attachment if it exceeds 20,000 characters.)

Reviewer #1: (No Response)

Reviewer #2: The manuscript has undergone significant revisions by the authors, resulting in notable improvements to its quality. Nevertheless, additional revisions are still necessary.

Abstract

Page 39, line 51: In the abstract and body of the manuscript, the authors utilized the term "HIV infection rate" to describe their objectives and inferential analyses. While the use of rate for descriptive analysis is acceptable, it is not appropriate for inferential statistics as the outcome was not aggregated. HIV status was at the individual level.

Page 39, line 57: "Data was..." should be revised to "Data were..."

Page 39, lines 60 and 61: Replace "LCA regressions..." with "Latent Class Regressions (LCR)." Also, make corresponding changes in the body of the manuscript.

Introduction

Page 41, lines 113 and 114: “In 2021, the number of new infections among adolescent girls and young women aged 15 to 24 decreased to an estimated 400,000." The authors should state the comparative year and the number of new infections.

Page 42, lines 169–171: The authors’ claim regarding the novelty of their study is inaccurate. Several studies have used LCA to identify barriers to health services.

https://bmchealthservres.biomedcentral.com/articles/10.1186/1472-6963-11-181

https://www.ncbi.nlm.nih.gov/pmc/articles/PMC10412727/

Methods

Page 45, line 294: “For children aged 0–9 years, the blood draw response rate was 68.5% for women and men”. Considering the age group, boys and girls are appropriate.

Page 46, line 324: Is there any rationale for limiting the analysis to those who only engage in vaginal sex, considering that young people participate in various high-risk behaviors such as anal sex, injection drug use, etc.? In line 335, it appears that NAIIS used only vaginal sex as a criterion to identify those who were sexually active. Are there any implications of narrowly defining the HIV risk within this population, particularly considering that injection drug use is becoming a public health challenge in Nigeria?

Discussion

In the manuscript, there are several occurrences, including Fig. 3, where the authors have utilized the term "prevalence" instead of "positivity." It is important to note that prevalence encompasses both old and new cases, whereas positivity rate specifically refers to the proportion of individuals who tested positive out of the total number of people tested.

7. PLOS authors have the option to publish the peer review history of their article (what does this mean?). If published, this will include your full peer review and any attached files.

If you choose “no”, your identity will remain anonymous, but your review may still be made public.

Reviewer #1: **Yes: **Luke Muschialli

Reviewer #2: **Yes: **Daniel A. Adeyinka

While revising your submission, please upload your figure files to the Preflight Analysis and Conversion Engine (PACE) digital diagnostic tool, https://pacev2.apexcovantage.com/. PACE helps ensure that figures meet PLOS requirements. To use PACE, you must first register as a user. Registration is free. Then, login and navigate to the UPLOAD tab, where you will find detailed instructions on how to use the tool. If you encounter any issues or have any questions when using PACE, please email PLOS at figures@plos.org. Please note that supporting information files do not need this step.

---

## [Author Response · Author response to Decision Letter 1]

24 Nov 2023

Dear Dr. Belayneh Mengist,

Editor,

PLOS ONE

23 November 2023 

Resubmission of manuscript “Latent class analysis of barriers to HIV testing services and associations with sexual behaviour and HIV status among adolescents and young adults in Nigeria”

Dear Dr Mengist, 

We are pleased to submit our revised manuscript entitled: “Latent class analysis of barriers to HIV testing services and associations with sexual behaviour and HIV status among adolescents and young adults in Nigeria” to the PLOS ONE for consideration for publication. We appreciate the careful review and constructive suggestions. 

We believe that the manuscript is substantially improved after addressing editor and reviewers’ comments. Submitted with this letter are the editor and reviewers’ comments with our responses in tabular form, including how and where the text was modified. 

The revision has been developed in consultation with all co-authors, and each author has seen and approved the final form of this revision. The funders had no role in study design, data collection and analysis, decision to publish, or preparation of the manuscript.

Thank you for your consideration of our revised paper.

Sincerely, 

Okikiolu Badejo for the co-authors

---

## [Decision Letter · Decision Letter 2]

3 Jan 2024

PONE-D-23-19074R2Latent class analysis of barriers to HIV testing services and associations with sexual behaviour and HIV status among adolescents and young adults in Nigeria

PLOS ONE

Dear Dr. Badejo,

Thank you for submitting your manuscript to PLOS ONE. After careful consideration, we feel that it has merit but does not fully meet PLOS ONE’s publication criteria as it currently stands. Therefore, we invite you to submit a revised version of the manuscript that addresses the points raised during the review process.

I would like to thank you for revising the manuscript. However, there are still points that need to be addressed.

First, please try to see the comments provided in R2 and address them. For example, you have been advised to use references appropriately (cited after a full stop). See line 75, 76, 81….224, 254….524.In the abstract, put Odds Ratio (OR) once; there is no need to put OR repeatedly. The same is true for 95% CI.You reported simply OR; it is not clear whether COR or AOR were reported. As long as a was regression adjusted for factors, AOR should be reported.Line 234: Citation is needed.Findings are reported from non-significant estimates. In line 51: OR 1.14, 95% CI 0.88-1.48, line 54; OR 1.24, 95% CI 0.98-1.57, OR 1.31, 95% CI 0.79-2.15 and line 55; OR 0.82, 95% CI 0.66-1.02. I know that non-significant findings can be reported as no association, but you generalized as having an association.In Table 4, you need to put variables adjusted for in the model as a footnote.When fitting models, multicollinearity should be considered. For example, in your model reported in Table 4, “place of residence” and “zone” might have a potential correlation. Please consider it.It is worthwhile to explain a bit more about how categories/subgroups (“low-risk perception, consent and proximity, testing site and cost and logistics”) were constructed. I recommend to clearly explain this in the method section.In Table 5, the “other” in the religion category is referenced; put as a footnote.It would be useful to conduct a subgroup analysis across sexes rather than putting it as a limitation.
Indicate which changes you require for acceptance versus which changes you recommendAddress any conflicts between the reviews so that it's clear which advice the authors should followProvide specific feedback from your evaluation of the manuscript

We look forward to receiving your revised manuscript.

Kind regards,

Belayneh Mengist, MPH

Academic Editor

PLOS ONE

Reviewers' comments:

Reviewer's Responses to Questions

**Comments to the Author**

1. If the authors have adequately addressed your comments raised in a previous round of review and you feel that this manuscript is now acceptable for publication, you may indicate that here to bypass the “Comments to the Author” section, enter your conflict of interest statement in the “Confidential to Editor” section, and submit your "Accept" recommendation.

Reviewer #1: All comments have been addressed

Reviewer #2: (No Response)

2. Is the manuscript technically sound, and do the data support the conclusions?

Reviewer #1: Yes

Reviewer #2: Yes

3. Has the statistical analysis been performed appropriately and rigorously? 

Reviewer #1: Yes

Reviewer #2: Yes

4. Have the authors made all data underlying the findings in their manuscript fully available?

Reviewer #1: Yes

Reviewer #2: Yes

5. Is the manuscript presented in an intelligible fashion and written in standard English?

Reviewer #1: Yes

Reviewer #2: Yes

6. Review Comments to the Author

Reviewer #1: Thank you very much for addressing these comments, the manuscript is now substantially improved and I can recommend it for publication. However, there is just one small change required.

Line 522 currently reads 'may exclude the perspective of sexual minorities or key population groups, transgenders, and other gender expansive individuals'. Transgenders is a slightly outdated term to use. Also, the important point I have been making in revisions about trans and gender expansive communities has not necessarily been about the sexual activity of the community as suggested in this most recent revision, but rather is about the exclusion of the group from the NAIIS survey as a whole. I therefore suggest the sentence is replaced with 'Moreover, the use of only vaginal sex as a measure of sexual activity in NAIIS excludes the experiences and activities of sexual minority groups, for example. In addition to this, the binary coding of sex in NAIIS excludes transgender and other gender expansive individuals from our analysis. These are significant communities in the context of the HIV epidemic in Nigeria and other settings (72,73), and future research should focus on sexual behaviours and the current HIV epidemic in these communities.'

Reviewer #2: Page 47, lines 283-285: The authors have used multiple imputations to treat missing data. To ensure reliability of their results, the authors should conduct sensitivity analysis and assess concordance or discordance of results from the imputed dataset and original dataset.

Page 47, lines 304-307: The authors wrote odd instead of odds.

Page 49, line 370: Table 2: As highlighted in my previous comments, p-value=0.0000 should be rewritten as p-value <0.001.

Page 52, The age groupings in other tables are not consistent with table 4, supplementary 2 and supplementary 4.

Supplementary 4; “correlates” was wrongly spelt.

7. PLOS authors have the option to publish the peer review history of their article (what does this mean?). If published, this will include your full peer review and any attached files.

Reviewer #1: **Yes: **Luke Muschialli

Reviewer #2: **Yes: **Daniel A. Adeyinka

---

## [Author Response · Author response to Decision Letter 2]

22 Jan 2024

Dear Dr. Belayneh Mengist,

Editor,

PLOS ONE

22 January 2024 

Resubmission of manuscript “Latent class analysis of barriers to HIV testing services and associations with sexual behaviour and HIV status among adolescents and young adults in Nigeria”

Dear Dr Mengist, 

We are pleased to submit our revised manuscript entitled: “Latent class analysis of barriers to HIV testing services and associations with sexual behaviour and HIV status among adolescents and young adults in Nigeria” to the PLOS ONE for consideration for publication. We appreciate the careful review and constructive suggestions. 

We believe that the manuscript is substantially improved after addressing editor and reviewers’ comments. Submitted with this letter are the editor and reviewers’ comments with our responses in tabular form, including how and where the text was modified. 

The revision has been developed in consultation with all co-authors, and each author has seen and approved the final form of this revision. The funders had no role in study design, data collection and analysis, decision to publish, or preparation of the manuscript.

Thank you for your consideration of our revised paper.

Sincerely, 

Okikiolu Badejo for the co-authors

---

## [Decision Letter · Decision Letter 3]

20 Feb 2024

PONE-D-23-19074R3Latent class analysis of barriers to HIV testing services and associations with sexual behaviour and HIV status among adolescents and young adults in NigeriaPLOS ONE

Dear Dr. Badejo,

Thank you for submitting your manuscript to PLOS ONE. After careful consideration, we feel that it has merit but does not fully meet PLOS ONE’s publication criteria as it currently stands. Therefore, we invite you to submit a revised version of the manuscript that addresses the points raised during the review process.

**Dear Dr. Okikiolu Badejo,**Thank you for addressing the comments. The manuscript is substantially improved and ready for publication. Some minor editorial comments.

There is some mixing of the method and result parts; for example, Figure 2 is the method section, and Line 185 to 188 is part of the result section.

Some minor editorial issues, for example, at line 330 and 435, need space “thefour”, “figuresin”

I recommend explaining the variables adjusted for in the multivariable models as a footnote.  

We look forward to receiving your revised manuscript.

Kind regards,

Belayneh Mengist, MPH

Academic Editor

PLOS ONE

Journal Requirements:

Reviewers' comments:

Reviewer's Responses to Questions

**Comments to the Author**

1. If the authors have adequately addressed your comments raised in a previous round of review and you feel that this manuscript is now acceptable for publication, you may indicate that here to bypass the “Comments to the Author” section, enter your conflict of interest statement in the “Confidential to Editor” section, and submit your "Accept" recommendation.

Reviewer #1: All comments have been addressed

Reviewer #2: All comments have been addressed

2. Is the manuscript technically sound, and do the data support the conclusions?

Reviewer #1: Yes

Reviewer #2: Yes

3. Has the statistical analysis been performed appropriately and rigorously? 

Reviewer #1: Yes

Reviewer #2: Yes

4. Have the authors made all data underlying the findings in their manuscript fully available?

Reviewer #1: Yes

Reviewer #2: Yes

5. Is the manuscript presented in an intelligible fashion and written in standard English?

Reviewer #1: Yes

Reviewer #2: Yes

6. Review Comments to the Author

Reviewer #1: Thank you for integrating all of my comments. Now that I have reviewed the final manuscript, I can now recommend for acceptance.

Reviewer #2: (No Response)

7. PLOS authors have the option to publish the peer review history of their article (what does this mean?). If published, this will include your full peer review and any attached files.

Reviewer #1: **Yes: **Luke Muschialli

Reviewer #2: **Yes: **Daniel A. Adeyinka

---

## [Author Response · Author response to Decision Letter 3]

21 Feb 2024

Dear Dr. Belayneh Mengist,

Editor,

PLOS ONE

21 February 2024 

Resubmission of manuscript “Latent class analysis of barriers to HIV testing services and associations with sexual behaviour and HIV status among adolescents and young adults in Nigeria”

Dear Dr Mengist, 

We are pleased to submit our revised manuscript entitled: “Latent class analysis of barriers to HIV testing services and associations with sexual behaviour and HIV status among adolescents and young adults in Nigeria” to the PLOS ONE for consideration for publication. We appreciate the careful review and constructive suggestions. 

We believe that the manuscript is substantially improved after addressing editor and reviewers’ comments. Submitted with this letter are the editor and reviewers’ comments with our responses in tabular form, including how and where the text was modified. 

The revision has been developed in consultation with all co-authors, and each author has seen and approved the final form of this revision. The funders had no role in study design, data collection and analysis, decision to publish, or preparation of the manuscript.

Thank you for your consideration of our revised paper.

Sincerely, 

Okikiolu Badejo for the co-authors

---

## [Editor Report · Decision Letter 4]

26 Feb 2024

Latent class analysis of barriers to HIV testing services and associations with sexual behaviour and HIV status among adolescents and young adults in Nigeria

PONE-D-23-19074R4

Dear Dr. Okikiolu Badejo,

We’re pleased to inform you that your manuscript has been judged scientifically suitable for publication and will be formally accepted for publication once it meets all outstanding technical requirements.

Kind regards,

Belayneh Mengist, MPH

Academic Editor

PLOS ONE
---

## [Editor Report · Acceptance letter]

27 Mar 2024

PONE-D-23-19074R4 

PLOS ONE

Dear Dr. Badejo, 

I'm pleased to inform you that your manuscript has been deemed suitable for publication in PLOS ONE. Congratulations! Your manuscript is now being handed over to our production team.

Kind regards, 

on behalf of

Mr Belayneh Mengist 

Academic Editor

PLOS ONE